# Accuracy of p16 IHC in Classifying HPV-Driven OPSCC in Different Populations

**DOI:** 10.3390/cancers15030656

**Published:** 2023-01-20

**Authors:** Roberto Gallus, Irene H Nauta, Linda Marklund, Davide Rizzo, Claudia Crescio, Luca Mureddu, Paolo Tropiano, Giovanni Delogu, Francesco Bussu

**Affiliations:** 1Otolaryngology, Mater Olbia Hospital, 07026 Olbia, Italy; 2Amsterdam UMC, Vrije Universiteit Amsterdam, Otolaryngology/Head and Neck Surgery, Cancer Center, 1081 Amsterdam, The Netherlands; 3Department of Clinical Science, Intervention and Technology, Department of Oto-Rhinolaryngology, Head and Neck Surgery, Karolinska University Hospital, Karolinska Institute, 17164 Stockholm, Sweden; 4Medical Unit Head Neck, Lung and Skin Cancer, Karolinska University Hospital, 17164 Stockholm, Sweden; 5Department of Surgical Sciences, Section of Otolaryngology and Head and Neck Surgery, Uppsala University, 75105 Uppsala, Sweden; 6Otolaryngology Division, Department of Medical, Surgical and Experimental Sciences, University of Sassari, Viale San Pietro, 43, 07100 Sassari, Italy; 7Otolaryngology Division, Azienda Ospedaliera Universitaria, 07100 Sassari, Italy; 8U.O.C. Otorinolaringoiatria, Università degli Studi di Cagliari, Policlinico Universitario Duilio Casula, 09042 Monserrato, Italy; 9Dipartimento di Scienze Biotecnologiche di Base, Cliniche Intensivologiche e Perioperatorie (Sezione di Microbiologia), Università Cattolica del Sacro Cuore, 00168 Roma, Italy; 10Mater Olbia Hospital, 07026 Olbia, Italy

**Keywords:** OPSCC, HPV-driven carcinogenesis, treatment de-intensification, HPV prevalence, diagnostic methods, specificity, false positives, molecular characterization, Western countries, developing countries

## Abstract

**Simple Summary:**

p16 IHC is the HPV detection method suggested by the current version of the TNM (AJCC 8th edition) for oropharyngeal squamocellular carcinoma. However, its reliability has been extensively discussed, and its applicability in every context, especially the enrollment of patients in de-intensification protocols, is debatable. Here, we discuss its limits, especially in populations with a low prevalence of HPV-driven oropharyngeal squamocellular carcinoma, and suggest the possible actions to be taken to overcome such limitations.

**Abstract:**

High-risk human papillomavirus (HPV) infection is a defined etiopathogenetic factor in oropharyngeal carcinogenesis with a clear prognostic value. The P16 IHC (immunohistochemistry) is a widely accepted marker for HPV-driven carcinogenesis in oropharyngeal squamous cell carcinoma (OPSCC); in the present paper, we discuss its reliability as a standalone marker in different populations. The literature suggests that rates of p16 IHC false positive results are inversely correlated with the prevalence of HPV-driven carcinogenesis in a population. We propose a formula that can calculate such a false positive rate while knowing the real prevalence of HPV-driven OPSCCs in a given population. As it has been demonstrated that p16 positive/HPV negative cases (i.e., false positives at p16 IHC) have the same prognosis as p16 negative OPSCC, we conclude that despite the valuable prognostic value of p16 IHC, relying only on a p16 IHC positive result to recommend treatment de-intensification could be risky. For this aim, confirmation with an HPV nucleic acid detection system, especially in areas with a low prevalence of HPV-related OPSCCs, should be pursued.

## 1. HPV as a Marker for Molecular Characterization in Oropharyngeal Squamous Cell Carcinoma

Evidence accumulated over the last 20 years [1,2,3,4,5,6] on the prognostic significance of high-risk Human Papillomavirus (hr-HPV) infection in oropharyngeal squamous cell carcinomas (OPSCCs) supported the inclusion of this parameter as the only acknowledged molecular marker in both the American Joint Committee on Cancer (AJCC) TNM (tumour, node, metastasis classification) and in main international guidelines for the head and neck [7]. We know that HPV + OPSCC cases show a better prognosis mainly because of their increased sensitivity to both cisplatin and radiotherapy [8,9,10,11], deriving at least in part from the fact that the lower (half) mutational rate in cancer cells keeps a wild type p53 gene, leaving the cells susceptible to pro-apoptotic agents [12]. Thus, the presence of HPV genes in the tumor tissue is associated with defined biological features and clinical behavior, and the fact that HPV-induced carcinogenesis can be easily diagnosed on small bioptic or cytological samples makes it a valid and important biomarker for OPSCCs [13]. AJCC recently included HPV-driven carcinogenesis as a decisive prognostic determinant, diversifying the TNM classification between HPV-driven and HPV-unrelated OPSCCs [14].

## 2. HPV in Relation to Treatment De-intensification

Since HPV-positive OPSCCs patients have markedly longer survival, the issue of long-term morbidity and quality of life (QoL) deterioration, deriving also from the aggressive multimodality treatments employed, becomes definitely more relevant [15,16]. The assumption that less aggressive treatments in HPV-positive OPSCCs could achieve the same oncological results with reduced long-term toxicity is the rationale underlying the perspective of treatment de-intensification, which has been evaluated in many, also ongoing clinical trials [17,18].

However, despite the fact that the concept of treatment de-intensification has become very popular with several studies hypothesizing treatment modulation according to HPV status in OPSCCs, the National Comprehensive Cancer Network (NCCN) justifies, at present, such an approach only in clinical trials [7]. On one hand, the NCCN panel is probably waiting for stronger evidence, coming from the ongoing and future clinical trials, to confirm that treatment de-intensification in HPV-driven OPSCCs is, for sure, beneficial as far as functional results are concerned and that it is also oncologically safe [19].

At any rate, we believe that the lack of consensus about the best diagnostic method(s) also contributes to hampering the safe introduction of HPV in head and neck clinical practice [20].

## 3. Detection Methods for HPV in HNSCC

The mere detection of HPV in a tumor sample does not imply a transcriptionally active virus, nor that the cancer is virus related. A diagnostic method that is to be utilized in clinical practice for the characterization of OPSCCs must detect a clinically relevant number of copies of transcriptionally active viral oncogenes. In fact, we need proof that HPV has impacted the carcinogenic process and currently contributes to the transformed phenotype (HPV-driven carcinogenesis), giving the tumor the typical clinical features of HPV-induced OPSCC. On the other hand, an assay detecting a small number of copies of the HPV DNA (which may come from a transient/not relevant infection or from contamination) may have a high rate of false positive results.

E6 and E7 appear to be the main drivers of cancerogenesis in HPV + HNSCC [6,21,22,23,24], while E5, another established oncogene, is usually not detected in OPSCC cancer cells. The expression of the E6 and E7 oncoproteins appears to be fundamental for the maintenance of the transformed phenotype [21]. Thus, the diagnosis of HPV-driven oropharyngeal carcinogenesis should demonstrate the expression of the E6 and E7 proteins, and in the absence of fully reliable immunohistochemical probes for the E6 and E7 proteins, methods detecting E6 and E7 mRNA in cancer cells are currently the gold standards for diagnosing an HPV-related HNSCC [20]. Unfortunately, this approach should be ideally carried out on fresh-frozen tumor samples that are not routinely available in standard clinical practices [25,26,27,28]. 

On the other hand, when it comes to methods that diagnose HPV-induced carcinogenesis in standard formalin-fixed paraffin-embedded (FFPE) samples, a gold standard is actually still missing [20]. Even if a plethora of methods are commercially available, and much more have been described in the scientific literature, practically all of them have been clinically validated in the uterine cervix but not in the head and neck. Current options for FFPE samples usually follow one or more of the following strategies, including the detection of viral DNA with or without a polymerase chain reaction (PCR), the detection of mRNA or DNA with in situ hybridization (ISH), and the detection of surrogate markers [29].

The detection of viral DNA with PCR is a non-quantitative method that is rather sensitive but poorly specific, as it tends to amplify contaminating material coming from non-relevant fragments of HPV DNA (non-cancerogenic strains or bystander infections) [28,30,31,32]. Such limitations in terms of specificity have been partially overcome by techniques that involve viral load quantification with RT-PCR and/or signal amplification methods (i.e., hybrid capture). Techniques based on signal amplification, in particular, have been shown to correlate nicely with mRNA detection methods in fresh frozen samples [27] and have been validated in cytological samples [33]. However, problems in terms of sensitivity may arise when the yield of the DNA extracted from the FFPE tissue samples is low or of poor quality [5]. Nonetheless, some landmark studies, including the original paper that demonstrated the role of hr-HPV infection in the cancerogenic process of a subset of OPSCC along with its prognostic relevance [1], and the work of Stransky et al. discussing the mutational pattern of HPV-related OPSCC [12], were indeed based on data coming from DNA-based techniques. 

ISH-based assays, despite being considered highly specific and having gained popularity, especially in the USA, share with other IHC-based techniques a certain dependance on an experienced histopathologist to correctly interpret the results, along with high costs and lengthy and complex procedures. This relative liability is probably behind the lack of diagnostic and prognostic reliability (particularly concerning sensitivity) and is sometimes lower than that of p16 IHC [4,25,26,30,34]. 

Immunohistochemistry for the p16 ^INK4A^ protein (p16 IHC) has been used since 2003 as a surrogate marker [35] and rapidly became a widely accepted method for diagnosing the HPV-driven OPSCCs thanks to its undeniable advantages, such as simplicity and low cost. However, p16 IHC is associated with many drawbacks and, in particular, a low specificity [36]. Despite these limitations, the expression of p16 is the criterion used for patient enrollment in many prospective trials of treatment de-intensification [28,37] and is acknowledged as valid in assessing HPV-related carcinogenesis in OPSCC, even by AJCC [14].

An alternative sequential strategy, including two highly sensitive methods, has already been validated by Dutch [30,31,38] and English [25] groups. Such a sequential strategy includes upfront p16 IHC, and when positive staining is observed, a subsequent HPV DNA detection with a PCR assay on FFPE samples for confirmation is performed. The Dutch authors report that the specificity of such an approach can be as high as 100%, thus drastically reducing the false positives coming from p16 IHC alone. 

Additionally, in Denmark and Sweden, combining HPV DNA and p16 is already recommended as essential for correct prognostication [39,40].

## 4. The Ominous Impact of False Positive p16 IHC

By analyzing the results of the Dutch and Swedish groups, it has been demonstrated that p16-positive OPSCCs that turn out to be HPV DNA-negative actually have the same prognosis [41] or slightly better [40] than p16-negative OPSCCs. These cases, if treated with a de-intensified regimen, are expected to have a markedly lower disease control, so treatment de-intensification would ultimately have a negative impact on oncological outcomes.

Therefore, it is obvious that the specificity, or more precisely, the minimal false positive rate, is a fundamental requirement for any assay used to recommend treatment de-intensification in supposedly HPV-driven OPSCC, raising concerns about the use of p16 IHC for this aim [20,42].

## 5. Factors Impacting the False Positive Rate of p16 IHC

In the literature, the rate of false positive cases with p16 IHC is extremely variable in the different series (some examples in Table 1).

However, for a given methodology used as a reference, the number of HPV-negative tumors overexpressing p16 tends to be constant. For example, when HPV-driven carcinogenesis is assessed with a combination of p16 IHC and PCR on genomic DNA, a relatively constant percentage (around 5.5% of HPV-negative tumors in a best-case scenario) overexpress p16 [5,41,42,45,47] (Table 1). Therefore, when using the sequential “Dutch” assay [30], the minimal proportion of HPV-negative/p16 + OPSCCs in a population can be predicted using the following formula: 5.5% × HPV non-driven rate. Likewise, the probability that a case that overexpresses p16 is HPV-negative (1-Specificity) can be easily predicted based on the prevalence of HPV-related carcinogenesis in that population with a simple formula (Figure 1). 

As shown in Table 2, the expected rate of false positives among p16 overexpressing cases, and that of the general OPSCC population, can be calculated to vary widely according to the prevalence of HPV-driven OPSCC in different populations. 

This means, for example, that about one-fourth of p16-positive OPSCCs will turn out to be HPV-negative in Sardinia compared to only 0.6% in Scandinavian countries. The total of 0.6% for a false positive rate is acceptable in the perspective of treatment de-intensification, but 25% is not because a high percentage of p16 positive cases would be undertreated with a relevant impact on prognosis. It means that p16 IHC may be deemed sufficient to recommend treatment de-intensification in Scandinavia or in the US, but not in Sardinia or in most of the world’s population, considering that in the most densely populated areas as India, Southern China, and Brazil a much smaller proportion of OPSCCs are HPV-driven. A graphical representation of what happens in a “best case scenario” (5.5% of HPV-negative tumors overexpress p16) can be seen in Figure 2 and Figure 3. Figure 2 shows how the probability that the p16 positive tumor is indeed HPV- changes according to the proportion of HPV-driven OPSCCs in a given population. The function is not linear and is expected to rise sharply in populations with an HPV-driven rate of OPSCCs below 25%. Notably, the probability is 1.8% in populations with 75% of HPV-driven OPSCCs, and it more than doubles (5.2%) in a population with an HPV-driven rate of 50%. Figure 3, starting from the same assumptions, shows how in this scenario, the number of false positive cases would rise constantly with a lower HPV-driven rate, with one misclassified patient every 100 tested at an 88% HPV-driven rate, two at 64%, three at 46% and four at 27% up to a maximum of 5.5 at 100%. 

## 6. Conclusions

To conclude, if we aim to safely implement treatment de-intensification for OPSCC, the above data and considerations must be kept in mind. We believe that the scientific community has two options in front of this evidence.

1. To accept p16 IHC as a standalone test only in populations with a high prevalence of HPV and defining a threshold (e.g., 50%, which would be associated with (1-specificity) = 5% in the best case scenario, according to the above formula for the prevalence of HPV-driven OPSCC, above which the expected rate of false positive p16IHC is considered acceptable.

2. To always confirm p16 IHC with hr-HPV nucleic acid detection before recommending treatment de-intensification in OPSCC. Notably, this is already the recommendation in Denmark and Sweden, even if these are countries with a notoriously high rate of HPV-induced OPSCCs.

## Figures and Tables

**Figure 1 cancers-15-00656-f001:**
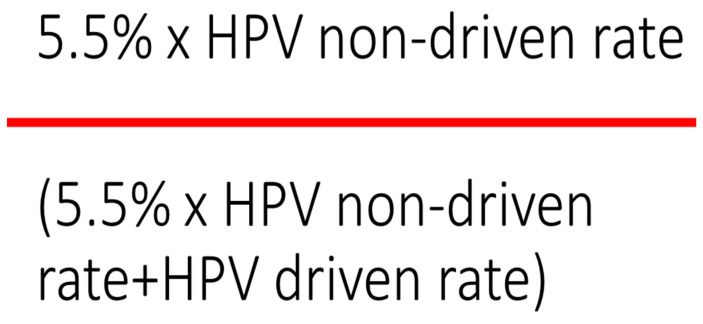
Formula to compute the rate of false positive p16 IHC if the reference is the combination between p16 IHC and PCR of genomic E6/E7.

**Figure 2 cancers-15-00656-f002:**
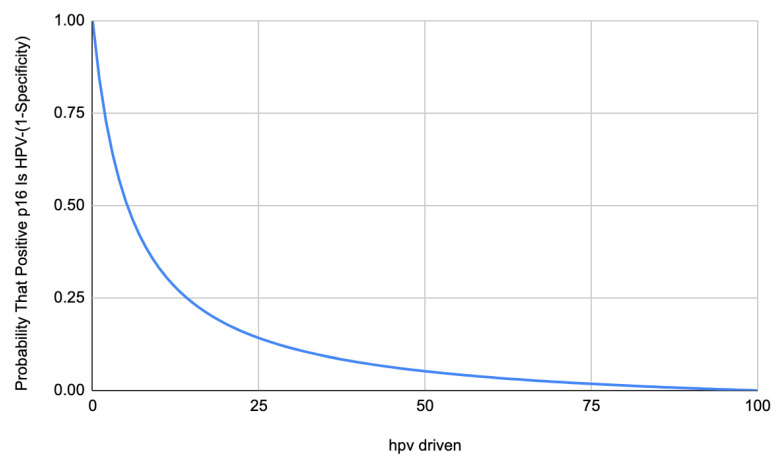
Graphical representation of the prediction of the rate of false positive tests (1-specificity) at p16 IHC when the reference is the combination of p16IHC + genomic DNA PCR, according to the proportion of HPV-driven OPSCCs in a given population.

**Figure 3 cancers-15-00656-f003:**
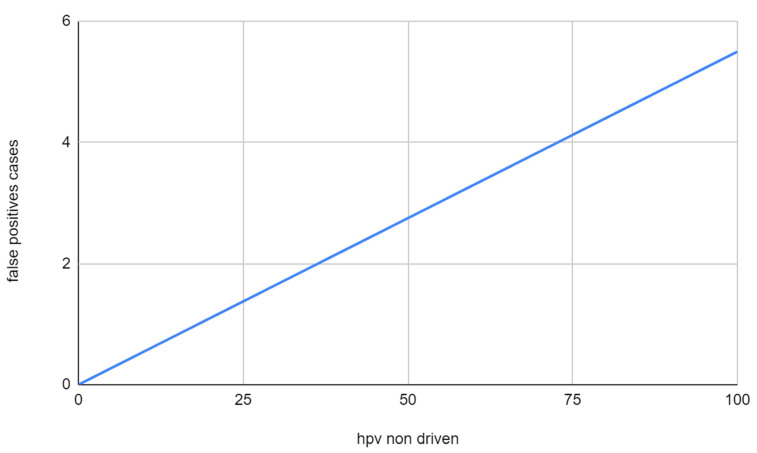
Graphical representation of the rate of false positive p16 IHC among OPSCC when the reference is the combination of p16IHC + genomic DNA PCR, according to the proportion of HPV non-driven OPSCCs in a given population.

**Table 1 cancers-15-00656-t001:** Accuracy of p16 IHC in different OPSCC series. Notably, for a certain assay, the FPR is relatively constant among very different populations (such as the Dutch, Germans, Czechs, and Sardinians), while specificity always decreases along with the HPV-driven rate.

Population	Rate of p16 Positive among HPV-Negative OPSCCs (FPR)	Rate of HPV-Induced OPSCC in the Population	Proportion of HPV-/p16 + OPSCC in the Population	Probability That Positive p16 Is HPV-(1-Specificity)	Detection Methods
Nauta (Holland) [41]	5.6%	28.2%	4%	12.3%	p16 (70%) GP5þ/6þ DNA PCR
Bussu (North Sardinia) [42]	5.7%	14.5%	4.8%	25%	p16 (70%) GP5þ/6þ DNA PCR
Bussu (Central Italy) [5]	20.6%	32%	14%	30.4%	p16 (70%) HPV E6 and E7 mRNA DNA (Hybrid Capture 2)
Saito (Japan) [43]	9.8%	32%	6.7%	17.2%	p16 (70%) DNA ISH DNA PCR E6
Benzerdjeb (France) [44]	13.6%	46.4%	7.2%	13.6%	p16 (70%) DNA ISH DNA PCR (CLART HPV2)
Schache (UK) [25]	11.3%	36.1%	7.2%	18%	p16 (70%) HPV16 E6 DNA qPCR HPV16 E6 RNA qPCR
Ang * (US) [4]	18.8%	63.8%	6.8%	10.3%	p16 (70%) ISH DNA
Rotnaglova (Czech Republic) [45]	5.3%	60%	2.2%	3.7%	p16 (50%) GP5þ/6þ DNA PCR
Linge (Germany) [46]	9.6%	21.7%	7.4%	25.6%	p16 (70%) GP5þ/6þ DNA PCR DNA PCR (LCD-Array HPV) HPV16 E6/E7 RNA
Prigge (Germany) [47]	5.7%	17.2%	4.7%	21.4%	p16 HPV16 E6 mRNA DNA (multiplex)
Oliva (Chile) [48]	26.3%	60.4%	10.4%	17.9%	p16 (70%) GP5þ/6þ DNA PCR
Méndez-Matías (Mexico) [49]	20%	39.2%	12.3%	23.7%	p16 (70%) DNA (INNO-Lipa)
Evans (UK) [50]	5.8%	50%	2.89%	5.5%	p16 (70%) GP5þ/6þ DNA PCR DNA ISH
Henneman (Netherlands) [51]	6.3%	34.9%	4.1%	10,5%	p16 (75%) DNA (INNO-Lipa)
D’Souza (USA) [52]	9.5%	55.8%	4.2%	7%	p16 (70%) DNA ISH RNA ISH
Lucas-Roxburgh (New Zealand) [53]	7,5%	62.6%	2.8%	4,3%	p16 (75%) DNA rtPCR
Ou (New Zealand) [54]	7%	74.5%	1.8%	2,3%	p16 (70%) GP5þ/6þ DNA PCR E6/E7 DNA rtPCR
Haeggblom (Sweden) [55]	15%	70%	4,5%	6%	p16 (75%) DNA (multiplex)

* HPV detected through FISH, with notorious sensitivity issues, with a relevant rate of false negative cases.

**Table 2 cancers-15-00656-t002:** Prediction of the rate of false positive tests (1-specificity) at p16 IHC when the reference is the combination of p16IHC + genomic DNA PCR, according to the proportion of HPV-driven OPSCCs in a given population. The calculation is made by the formula in Figure 1.

Rate of HPV-Induced OPSCC	Expected Rate of False Positive among p16 Overexpressing OPSCC	Expected Rate of False Positives in the Whole OPSCC Population
28% (Holland)	12.4%	3.9%
14.5% (North Sardinia)	25%	4.7%
80% (US)	1.3%	1.1%
90% (Scandinavian Countries)	0.6%	0.5%

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
