# Peer review of "Accuracy of p16 IHC in Classifying HPV-Driven OPSCC in Different Populations"

_cancers, 2023, doi:10.3390/cancers15030656_

Round 1
Reviewer 1 Report
Gallus et al. have summarized the determination of HPV-associated oropharyngeal cancer. They have described the algorithm of the diagnostic approach of p16 IHC detection followed by HPV PCR. Thereby, the subgroup of p16 positive and HPV negative patients have an inferior survival which is comparable to p16 negative patients. Therefore, deintensivation of the therapy should not be performed in this subgroup. The authors describe this fact very nicely and use actual literature. The focus on this subgroup is very important and needs to be addressed especially because in diagnostic routine only p16 determination is required. Additionally, the size of this percentual proportion of this subgroup (p16 positive/HPV negative) depends on the prevalence of the p16 prevalence.
The manuscript is well written and this subgroup needs to be addressed, however the title is misleading and should be revised.
Author Response
Gallus et al. have summarized the determination of HPV-associated oropharyngeal cancer. They have described the algorithm of the diagnostic approach of p16 IHC detection followed by HPV PCR. Thereby, the subgroup of p16 positive and HPV negative patients have an inferior survival which is comparable to p16 negative patients. Therefore, deintensivation of the therapy should not be performed in this subgroup. The authors describe this fact very nicely and use actual literature. The focus on this subgroup is very important and needs to be addressed especially because in diagnostic routine only p16 determination is required. Additionally, the size of this percentual proportion of this subgroup (p16 positive/HPV negative) depends on the prevalence of the p16 prevalence.
The manuscript is well written and this subgroup needs to be addressed, however the title is misleading and should be revised.
- Thank you, we changed the title to better reflect the manuscript content

Reviewer 2 Report
The present paper aims to discuss the reliability of p16 IHC, a widely used assay for the diagnosis of HPV-driven carcinogenesis, as a standalone surrogate marker of HPV status. The paper is discussing an important topic in our field, however, the study design and type are not clear “the paper seems to be a letter to the editor or a narrative review”. It is highly recommended to define clearly the type of paper in the title, abstract, and manuscript.
It is not clear how the authors obtain the data to construct Figures 2 and 3. It is recommended to include a description in the manuscript of how the authors got the data. In addition, based on these two graphs (figures 1 and 2), what did the authors observe? this is not clear, an answer to this question should be addressed in the manuscript.
Some abbreviations are not defined at the first use in the abstract and the manuscript; including p16 IHC (page 1, line 25), HPV (page 1, line 25), TNM (page 1, line 26), AJCC (page 1, line 26), NCCN (page 2, line 53), and TNM (page 3, line 62).
Author Response
The present paper aims to discuss the reliability of p16 IHC, a widely used assay for the diagnosis of HPV-driven carcinogenesis, as a standalone surrogate marker of HPV status. The paper is discussing an important topic in our field, however, the study design and type are not clear “the paper seems to be a letter to the editor or a narrative review”. It is highly recommended to define clearly the type of paper in the title, abstract, and manuscript.
- Thank you, we clarified the article type with the Editor (communication) and modified it according to the journal guidelines for this type of article. We also clarified that in the article.
It is not clear how the authors obtain the data to construct Figures 2 and 3. It is recommended to include a description in the manuscript of how the authors got the data. In addition, based on these two graphs (figures 1 and 2), what did the authors observe? this is not clear, an answer to this question should be addressed in the manuscript.
- Thank you, we clarified how we constructed the figures and their impact and interpretation.
Some abbreviations are not defined at the first use in the abstract and the manuscript; including p16 IHC (page 1, line 25), HPV (page 1, line 25), TNM (page 1, line 26), AJCC (page 1, line 26), NCCN (page 2, line 53), and TNM (page 3, line 62).
- Thank you, we checked and modified the manuscript to be sure we defined each abbreviation at its first occurrence.
